# A Decade of Therapeutic Challenges in Synchronous Gynecological Cancers from the Bucharest Oncological Institute

**DOI:** 10.3390/diagnostics13122069

**Published:** 2023-06-15

**Authors:** Laurentiu Simion, Elena Chitoran, Ciprian Cirimbei, Daniela-Cristina Stefan, Ariana Neicu, Bogdan Tanase, Sinziana Octavia Ionescu, Dan Cristian Luca, Laurentia Gales, Adelina Silvana Gheorghe, Dana Lucia Stanculeanu, Vlad Rotaru

**Affiliations:** 1Carol Davila University of Medicine and Pharmacy, 050474 Bucharest, Romania; lasimion@yahoo.com (L.S.); cristinastefan10@gmail.com (D.-C.S.); luca_dan94@yahoo.com (D.C.L.); laurentia.gales@yahoo.com (L.G.); adelina-silvana.gheorghe@drd.umfcd.ro (A.S.G.); dana.stanculeanu@umfcd.ro (D.L.S.); rotaru.vlad@gmail.com (V.R.); 2Department of Oncological Surgery I, Bucharest Oncological Institute, 022328 Bucharest, Romania; 3Department of Pathology, Bucharest Oncological Institute, 022328 Bucharest, Romania; ariana.neicu@gmail.com; 4Department of Thoracic Surgery, Bucharest Oncological Institute, 022328 Bucharest, Romania; bogdansen@gmail.com; 5Department of Medical Oncology, Bucharest Oncological Institute, 022328 Bucharest, Romania

**Keywords:** synchronous primary gynecological cancer, synchronous cancers, ovarian cancer, endometrial cancer, endometrioid type, adjuvant treatment, pelvic exenterations, multiple primary malignancies, HPV-related cancers

## Abstract

The aim of our study is to present the particularities of a specific subset of gynecological cancer patients in Romania. We present a review of synchronous gynecological neoplasia (SGN) treated in the Bucharest Oncological Institute’s surgery departments over a decade. Between 2012 and 2022, 7419 female patients with genital malignancies were treated. We identified 36 patients with invasive synchronous primary gynecological cancers (0.5%) and 12 cases with one primary gynecological and another primary invasive pelvic cancer (rectal/bladder). All recurrent, metastatic, or metachronous tumors detected were excluded. Demographic data, personal history, presenting symptoms, pathologic findings, staging, treatment, and evolution for each case were recorded. Usually, the most common SGN association is between ovarian and endometrial cancer of endometrioid differentiation (low-grade malignancies with very good prognosis). However, we noticed that, given the particularities of the Romanian medical system, the most frequent association is between cervical and endometrial, followed by cervical and ovarian cancers. Moreover, the cancer stage at diagnosis is more advanced. In countries with low HPV vaccination rate and low adherence to screening programs, SGNs can present as extremely advanced cases and require extensive surgery (such as pelvic exenterations) to achieve radicality. This multimodal treatment in advanced cases with high tumor burden determines a reduction in survival, time until progression, and quality of life.

## 1. Introduction

Synchronous primary invasive tumors of the female genital tract are not a very common occurrence, yet when they arise, they can lead to serious challenges when it comes to complete diagnosis and treatment. The coexistence of multiple gynecological malignancies was reported in the early literature between 1 and 6% [1,2,3]. However, in more recent publications, a reduction in the incidence to 0.7–1.8% of patients treated for genital cancers is described [4]. The most common synchronous gynecological neoplasms (SGNs) are reported to be ovarian and endometrial cancers, this association being reported to occur in up to 10% of all women with ovarian cancer and in 5% of women with endometrial neoplasia [5,6]. Although SGNs are usually credited with low-grade malignant behavior, the authors consider that this statement cannot be taken as a general rule, as different national particularities, such as status of HPV vaccination, medical education of general population, and accessibility to medical services, influence the overall prognosis.

Diagnosis of synchronous gynecological malignancies can be challenging as symptoms may be non-specific and can overlap with those of other gynecological conditions. A combination of imaging studies, laboratory tests, and biopsy procedures may be used to confirm the diagnosis [4].

The approach to treatment recommendations in synchronous gynecological malignancies is not always clear and there is currently no standard of care defined. These kinds of situations serve as a reminder of the value of in-depth discussions, proper consent procedures, multidisciplinary approach and collaboration, and patient participation in treatment decision-making. Moreover, the choice of treatment depends on factors such as the stage and type of cancers, the patient’s overall performance status, and the potential side effects of the treatments [4,5,6].

Therefore, there is a need for updated guidelines, real-world data, and shared clinical experience from the centers with expertise in treating SGNs.

## 2. Materials and Methods

We performed an extensive review of the internal database of our institution, selecting all female patients that have been treated in the surgery departments of the “Prof. Dr. Al. Trestioreanu” Oncological Institute, Bucharest, Romania for primary invasive gynecological cancer over a ten-year period between 2012 and 2022.

We selected all patients with SGNs for enlistment in our study. The diagnostic criterion was the detection of different histological types upon pathologic evaluation of the resection specimens. Exclusion criteria were metachronous tumors and local invasion of the tumor of another genital tract organ. Metastatic or recurrent cases were also excluded. We also excluded patients with less than 6-month follow-up. We defined as primary neoplastic disease the one that was symptomatic and made the patient seek out medical assistance. The secondary cancer was considered the one with less symptomatology and usually diagnosed from the surgical resection specimen upon microscopic analysis.

All the cases identified were studied further in order to identify particularities about their histopathological type and clinical stage, symptoms at presentation, diagnostic, treatment, and prognosis. We collected demographic data for each patient including age at the time of diagnosis, hormonal usage, data about pregnancies and menopausal status, and data about risk factors and exposure to known carcinogens. Initial symptoms were recorded. Complete staging was performed in all cases included in our study, using FIGO classification. The treatment involved the adequate surgical procedure for each individual patient, and adjuvant therapy was administered if indicated. Further data collected aimed to describe the evolution of each case—follow-up duration, existence of recurrence, and disease-free period until recurrence; the overall survival of each case was recorded (this was measured in months starting from the date of diagnosis and until death or most recent follow-up).

## 3. Results

Between 2012 and 2022, a total of 7419 women were treated in the surgery departments of the Oncological Institute “Prof. Dr. Al. Trestioreanu” from Bucharest for primary invasive gynecological cancers: 1092 women with ovarian cancer, 1690 with endometrial cancer, 4288 with cervical neoplasia, and 349 with other types of genital neoplasia (vulvar, vaginal or tubar).

From these 7419 cases, we selected patients diagnosed with synchronous primary invasive cancers. We identified 36 patients with invasive SGN, representing 0.49% of the total population, and 12 patients that presented with a primary gynecological neoplasia synchronous with another pelvic primary invasive cancer (rectal or urinary bladder)—0.17%. We also identified 67 cases presenting with an invasive form of gynecological neoplasia and an intraepithelial cancer of another organ of the feminine genital tract—0.9%. Figure 1 presents the incidence of the synchronous genital cancers in our study group.

In the 36 patients with SGNs, the most frequent neoplastic synchronicity was the association of endometrial and cervical cancers (12 cases—33.3%), followed by the association of ovarian and cervical cancers (10 cases—27.8%), the commonality being cervical cancer. The most frequent association described in the literature (between endometrial and ovarian cancer) was, in our case, only third in frequency (six cases—16.7%). This discrepancy can be easily explained if we take into account that our institution is responsible for the national screening program for cervical cancer for the entire South-Eastern part of Romania; thus, the number of cervical cancer cases diagnosed and treated in our institution is higher than the median national level, representing around 25% of our patient load. Additionally, in Romania, the rate of HPV vaccination is extremely low (12–13% of eligible population); thus, the decrease in incidence of cervical cancer that was registered in other countries over the last decade (cervical cancer not figuring among the first five female neoplasias anymore) was not followed by the same trend in our country. The low acceptance of HPV vaccination in Romania is due to low general medical education among the populace, fueled by the anti-vaccination movement on social media and also ethnic and social beliefs. However, the exact incidence of cervical cancer is uncertain because of the lack of a complete national cancer registry. We continue to diagnose an increased number of cervical cancer cases, some of them in advanced stages (including stage IVa—locally invasive). None of our patients were ever vaccinated against HPV. Not all patients were tested for carcinogenic strains of HPV and an exact prevalence of the infection in the study group is unknown, but 23 of the patients were tested with 18 positive results. Meanwhile, although the incidence of endometrial cancer has risen in recent years and is currently above the American and European median value, only 8% of all our female patients were affected by this disease [7,8].

The mean age at diagnosis was 56.6 years (with ranges between 34 and 79 years). Compared to other series published in literature, our group consists of older women (Figure 2). Unlike the standard series reported before, we observed a preponderance of cervical cancer cases, which usually affect older women.

Median parity was 1.3 births (with ranges between zero and five births). Postmenopausal status was registered in 29 patients (80.5%). There were no cases that received hormonal substitution. Only two patients, aged 34 and 40, had a history of hormonal birth-control pills.

Initial symptoms included abnormal vaginal/uterine bleeding (80%), abdominal distension (25%), clinical adnexal mass (17%), pelvic pain (17%), and other symptoms (14%).

The most frequent histopathological type of cervical cancer associated with SGNs in our group was non-keratinized squamous cell carcinoma (*n* = 20). This histopathological diagnosis applied for both the cases in which the cervical carcinoma was the primary cancer and also in the cases in which the cervical cancer was diagnosed after radical surgery for ovarian/endometrial cancer upon histopathological evaluation of the resection specimen. 

Unlike endometrial endometrioid type carcinoma, the non-keratinized squamous cell type is characterized by a worse evolution. When the cervical cancer is diagnosed after surgery for ovarian cancer, the stage of the primary ovarian cancer is usually advanced (most of our cases presenting with large adnexal masses and even peritoneal nodules—stages II/III). By contrast, the cases associated with endometrial cancers have a better prognosis due to the fact that the abnormal bleeding forces the patients to seek out medical assistance sooner and thus be diagnosed sooner.

Table 1 presents the principal histopathological characteristics of both the primary and secondary cancers in our patients. Table 2 summarizes the initial pathologic stadialization of the SGN cases in our study group.

The treatment involved the adequate surgical procedure for each case, and adjuvant therapy was administered if it was indicated. In our study group, all patients underwent radical intent surgery. In the case of patients with SGNs involving uterine cancers with endometrioid differentiation, a radical lymphadenocolpohysterectomy was usually sufficient; some of our patients required more extensive surgery, such as multiple organ resection (in stage IIIc ovarian cancer) or pelvic exenterations (three of our patients required such interventions as a primary curative procedure) followed by regional lymphadenectomy.

The median follow-up duration was 64 months (range: 6–115 months). The 5-year survival rate in our group was 55.5%, and the disease-free survival (DFS) was 60.2 months. We performed a non-linear analysis of survival over time taking into account the observational loss, and the Kaplan–Meier curve obtained is presented in Figure 3. According to different authors, the 5-year survival rate in patients with the synchronous gynecological cancers is 71–96% [9,10,11,12,13]. By contrast, our cohort has a lower 5-year survival rate of only 55.5%, which in our opinion is due to the advance stage at diagnosis of some of our cases.

In our study group, we had 10 recurrences (seven in the ovarian cancer cases and three in the cervical cancer cases). Recurrences were diagnosed through vaginal cytology/biopsy or imaging techniques combined with tumoral markers for abdominal recurrences. The recurrence occurred after a period free of disease between 6 and 28 months. Five of our patients developed metastatic disease—two with endometrial cancers developed peritoneal carcinomatosis, one case with cervical cancer developed a metastatic supraclavicular lymphadenopathy, and two cases with ovarian cancer developed pleura-pulmonary metastatic disease.

Among our patients with recurrent disease, we had two patients that underwent pelvic exenterations (or pelvectomies). Although credited with a high procedure-associated morbidity (over 40% major morbidity for standard techniques and up to 50% when the exenteration is extended to include lateral pelvic structures such as iliac vessels or bony parts of the pelvis), the intraoperatory and immediate postoperative mortality approaches 0, so this procedure should be considered in both radical and palliative settings for selected cases and after extensive discussions with the patients in order to ensure they have realistic expectations and understand the consequences of this surgical technique. Pelvectomies performed on our patients with SGNs (*n* = 5, three as primary radical surgery and two for recurrences) had 0 mortality and a 40% morbidity rate (two cases). The major morbidity required reintervention in two cases: one for “empty pelvis syndrome” and secondary bowel obstruction, and one for acute lower limb ischemic syndrome.

The quality of life perceived by the patients that undergo a pelvic exenteration is diminished due to the discomfort caused by urinary or digestive stomas (time and effort needed to take care of the stoma). The presence of a definitive stoma is associated with a profound psychological impact: most patients develop a form of depression, sometimes severe, caused by the alteration of self-image that disrupts daily activities and social interactions (including family life) and gives rise to a need to conceal the stoma from others. However, when asked about symptoms they experienced prior to surgery (such as abdominal discomfort, nausea, constipation, back pain, urinary discomfort, and so on), the patients reported an overall improvement.

When pelvic exenterations involve vascular structures or are performed on patients with rich vascular surgical history, these procedures have a higher degree of morbidity.

In our group, we observed one case that required a lateral extension of the standard technique of posterior pelvectomy with external iliac vessels resection for recurrent cervical cancer 15 months after a radical lymphadenocolpohysterectomy for cervical and ovarian synchronous neoplasia. We used a synthetic graft reconstruction technique for the iliac artery. The vein was not reconstructed due to the fact that it was thrombosed prior to surgery and the collateral circulation already was in place. This is most often the case in patients needing pelvic exenterations with iliac vessels resection. The patient had no vascular complications after surgery.

We also had one patient with ovarian and cervical neoplastic synchronicity that had numerous prior vascular surgical procedures: infrarenal aortic thrombectomy for aorto-iliac disease, left aorto-femoral bypass with Dacron prosthesis, right deep femoral bypass on Linton patch with silver Dacron prosthesis, and proximal prosthetic–popliteal extension with a reinforced conical poly-tetra-flour-ethylene (PTFE) prosthesis. She underwent a radical anterior pelvectomy for stage IVa cervical cancer diagnosed by biopsy (urinary bladder invasion diagnosed by cystoscopy) and a large cystic adnexal mass associated with elevated ovarian marker (CA-125). Due to the voluminous adnexal mass which was considered a contraindication of neoadjuvant radiotherapy, the patient was first referred to the surgical department. The patient suffered no intraoperatory complications. However, on the third day after the procedure, she developed acute left lower limb ischemia, which required a vascular reintervention. We performed the removal of the thrombus in the prosthetic left branch of the deep femoral artery and of the popliteal-tibial axis via the proximal popliteal pathway using a Fogarty type probe, followed by prosthetic–proximal popliteal bypass with collagenized Dacron prosthesis no. 8. The ulterior evolution was uneventful, with the remission of the peripheral ischemic syndrome. The synchronous ovarian cancer was diagnosed on the resection specimen upon histopathological examination. Figure 4 presents aspects of this case.

We registered 15 patients dying during the follow-up period (nine due to malignant recurrence, one due to other cancers—breast cancer, and five due to non-malignant pathologies (cardiac insufficiency, chronic renal failure, thrombembolic events)).

## 4. Discussion

The etiology of multiple invasive primary cancers (MIPCs) of the female genital tract is unclear. One of the first theories trying to explain MIPC etiology is the “cancerization field”, first described by Slaughter in 1953, when he studied the presence of histological abnormal tissue surrounding oral squamous cell carcinoma [14]. This theory tried to explain the development of multiple primary tumors and locally recurrent cancer. Slaughter explains the occurrence of synchronous or recurrent cancer by the appearance of pre-neoplastic tissue islands as a result of the accumulation of genetic errors in cell replication under the influence of repeated exposure of mucosal membranes to common carcinogens and the subsequent selection of a mutant cellular clone that will gain a proliferative advantage over normal cells. Later, other genetic errors accumulate, which will lead to the expansion of the “cancerization field”, slowly replacing normal mucosa, thus favoring the appearance of synchronous or recurrent cancer. After Slaughter’s initial description, similar descriptions of the “cancerization field” were made for other head and neck cancers (oral cavity, oropharynx, and larynx) and for cancers of the lung, vulva, esophagus, cervix, ovary, breast, skin, colon, and bladder [15,16].

Another theory that can explain the etiology of MIPC of the female genital tract is the common origin of the epitheliums (the SGNs develop from different sites with similar histoembriological origin). Structures lined by Müllerian epithelium can frequently be seen outside the uterus and are called the “secondary Müllerian system”. This epithelium has an extremely high metaplastic potential and is involved in the etiology of many pathological entities, ranging from benign endosalpingiosis to highly malignant ovarian tumors [17,18]. The theory of the “secondary Müllerian system” proposed that the epithelia of cervix, uterus, fallopian tubes, ovaries, and peritoneal surface have shared molecular receptors and respond to the same carcinogenic stimulus, leading to the development of multiple primary malignancies of similar histology synchronously [19,20,21].

There are also studies that have shown the relation between SGNs and Lynch syndrome (Hereditary Non-Polyposis Colon Cancer). Lynch syndrome is a hereditary disease characterized by constitutive mutations in the DNA mismatch repair pathway (most commonly MLH1, MLH2, MSH6, or PMS2 genes). The accumulation of replication errors leads to the selection of mutant clonal cells, a precursor to cancers. This genetic syndrome is known for multiple cancers of the colon and endometrium. Lifetime risk for developing endometrial cancer in patients with Lynch syndrome is 39%. Since 1–3% of all endometrial cancers are connected to Lynch syndrome, it has been suggested that universal screening by immunohistochemical staining for the four most common altered genes (MLH1, MLH2, MSH6, or PMS2) is a method for identifying previously unknown hereditary cases and therefore provides an opportunity for appropriate counseling to the patients and their families [22]. However, the association between Lynch Syndrome and SGNs including endometrial and ovarian cancers is not clearly understood [23].

The incidence of MIPC is rising due to the increased overall survival of oncological patients (continuous advances in medical technology and procedures) and continuous exposure to an ever-increasing number of carcinogens.

There are four types of MIPC: multicentric tumors (different tumors of the same organ), systemic multiple tumors (different tumors of two or more anatomically and functionally related organs), MIPC of paired-organs (such as breast or kidney), and random MIPC (different tumors of unrelated organs). SGNs are part of the systemic MIPCs. Among the SGNs, the most common neoplastic association is ovarian and endometrial cancers [24,25,26,27,28], most frequently with endometrioid histotype [29,30]. However, as we saw in our results, this is not a general rule and can be influenced by external factors pertaining to the medical system of a country, such as accessibility to medical services, general medical education of the population, screening programs available, and also the status of HPV vaccination. The high incidence of cervical cancer in Romania is consequent to low adherence to screening programs of eligible women, lack of parental approval for HPV vaccination, as well as a general lack of education regarding HPV infection and the severity of its consequences [31]. The incidence of HPV infection among women aged 18–59 years is 40% for all HPV types and 20% for high-risk HPV types [32]. Considering the fact that Romania is second in Europe when it comes to number of deaths from cervical cancer, it can be argued that there is a major lack of education about HPV infection, its role in cancer etiology, and prevention through HPV vaccination [33]. In Romania every year, 3380 women are diagnosed with cervical cancer, and 1805 die from this disease [34,35,36]. The same factors explain the atypical component of our study group (SGNs involving cervical cancers are more frequent than usual) and the atypical results when it comes to survival rates and DFS (some of our cervical cancer cases were diagnosed in extremely advanced stages and required significant surgical procedures which were followed by procedure-related morbidity and mortality, thus altering the overall results).

The first challenge raised by SGNs is differentiating between metastatic disease in a secondary organ of a unique primary tumor and true synchronous tumors. The diagnosis criteria for SGNs are detection of distinct histologic subtypes, or all of the following rules (if histological subtypes are similar): both tumors are confined to primary sites with no direct extension between tumors; no lymphovascular tumor emboli; no or only superficial myometrial invasion; and no distant metastasis [37].

This differentiation is essential because SGNs are usually diagnosed in earlier stages [38] and have a better prognosis than metastatic disease [39]. Reported clinical–pathological features of dual primary cancer also include younger age and histological lower grade. Still, they require the adaptation of the therapeutic strategy to the particularities of each case. Most of them need a surgery-first approach followed by adjuvant therapy, as indicated. A perfect exemplification of this aspect is the fact that, in our study group, the cases presenting with biopsied cervical cancer and adnexal masses were first deferred to the surgical departments rather than undergoing radiotherapy for the already diagnosed cancer. The radical total abdominal hysterectomy with bilateral salpingo–oophorectomy performed in such cases served a dual role, both as a diagnostic tool (the secondary cancer being diagnosed on the resection piece) and as a treatment option (resolving the primary tumors and guiding ulterior chemotherapy and facilitating adjuvant radiotherapy).

The endometrioid differentiation of tumors involved in SGNs is accompanied by better prognosis than other histotypes [40]. These are considered Type I endometrial cancers [41] according to the categories identified through epidemiological studies by Bokhman, with endometrioid carcinoma corresponding to Type I and serous carcinoma to Type II [42]. Moreover, it has been suggested that the SGN consisting of endometrioid-type tumors of the ovary and endometrium have a better overall prognosis than endometrioid ovarian cancer alone due to the fact that the synchronous ovarian tumor is usually diagnosed in stage I [43]. The synchronicity between ovarian and uterine cancers is credited with better prognosis than ovarian cancer alone, regardless of histotype, and for this type of SGN, surgery alone may be enough treatment for stage I cases [10]. Synchronous Stage I ovarian carcinoma has no impact on disease-free, overall or cancer-specific overall survival in univariate and multivariate analyses in cases with endometrioid endometrial cancer [44]. In stage I cases, intensive follow-up protocols do not change the overall survival outcome [45].

The endometrioid-type carcinoma’s immunohistochemical markers are intense positivity for cytokeratin and epithelial membrane antigen, cytokeratin 7 (CK-7), lack of staining for cytokeratin 20, positive basal or perinuclear staining for vimentin, and membranous and nuclear presence of beta-catenin staining. There can be focal positivity for chromogranin, synaptophysin, and CD56. Low-grade endometrioid carcinomas show diffuse, very strong positivity for estrogen and progesterone receptors. The positivity of estrogen or progesterone receptors (over 90%) is significantly higher in type 1 compared to that in type 2 endometrial cancer (71% or 64%) in both premenopausal and postmenopausal women [46]. Overexpression of p53 and p16 appears in 30% of high-grade endometrioid carcinomas.

The last but not least of the challenges raised by SGNs is the fact that, most often, they are not diagnosed prior to surgery by clinical examination and imaging techniques. However, we need to consider the possibility of synchronous gynecological tumors when confronted with any primary gynecological cancer, as a diagnosis of SGNs can alter both the treatment plan and prognosis of the patient. The association can adjust the treatment plan by changing the sequence of the multimodal treatments (usually surgery takes a primary role and it is the first one used, due to the fact that radiotherapy for cervical/endometrial cancer can be hindered by the volume of a synchronous adnexal mass). Usually, the secondary cancer is diagnosed upon histopathological analysis of the surgical specimen, and as a result, the subsequent adjuvant chemotherapy is modified.

### Limitations

This study has several limitations, including that it used a retrospective review, which limits long term follow-up. The relatively low number of cases limits the power of generalization of the study. The socioeconomic conditions, particular for a country with an average level of development, did not allow the performance of tests or investigations (genetic tests, PET CT scan imaging, biological tests, etc.). 

However, our study reflects an extremely current problem, investigated in depth over a significant period of time in the largest oncology institute in Romania. Considering the local socioeconomic conditions, they are not specific only to Romania, so that at least in low- and middle-income countries this type of pathology is similarly encountered.

Another limitation of our study is the fact that we did not use a Modified Frailty Index (mFI), which was suggested as a tool for definition of frail patients, frailty being an important factor in decreasing disease-free survival rates and overall survival and increasing postoperative complications, hospital stay length, and Intensive Care Unit admission [47].

## 5. Conclusions

SGNs are very rare entities that should be considered when confronted with cases diagnosed with a cervical cancer and also presenting abnormal uterine bleeding and/or adnexal masses. The neoplastic synchronicity raises several diagnostic challenges and requires the adaptation of therapeutic protocols to the specifics of each case.

Although usually described in the literature as low-grade malignancies with favorable prognosis, in the context of countries with low HPV vaccination rate and high incidence of female genital tract cancers, SGNs can present as extremely advanced cases and will require extensive surgery (such as pelvic exenterations) in order to achieve radicality. We consider that it is paramount to strive for achieving a better adherence to screening programs and to HPV-vaccination programs in Romania in order to reduce cervical cancer incidence and alleviate the incidence and prognosis of SGNs by decreasing the morbidity and mortality associated to cervical cancer.

## Figures and Tables

**Figure 1 diagnostics-13-02069-f001:**
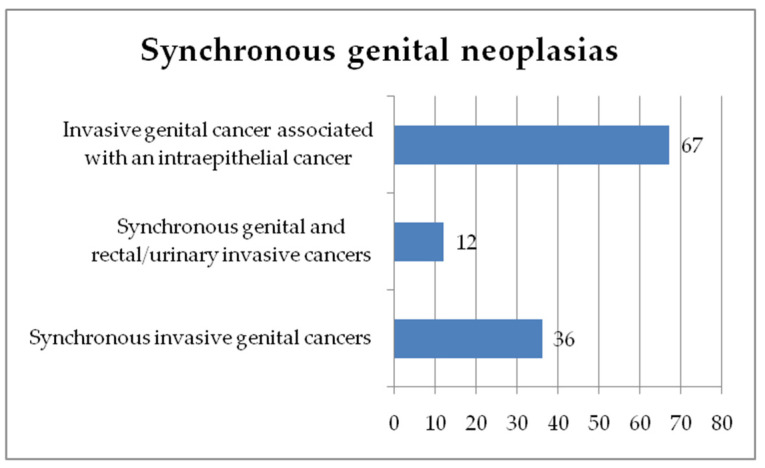
Incidence of the synchronous genital cancers in our study group.

**Figure 2 diagnostics-13-02069-f002:**
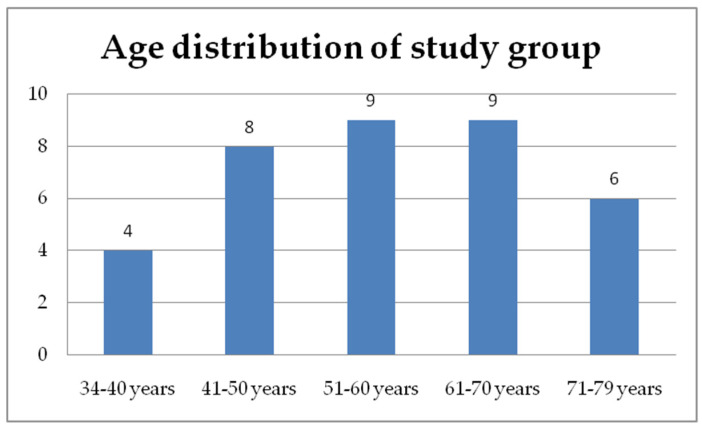
Age distribution of study group.

**Figure 3 diagnostics-13-02069-f003:**
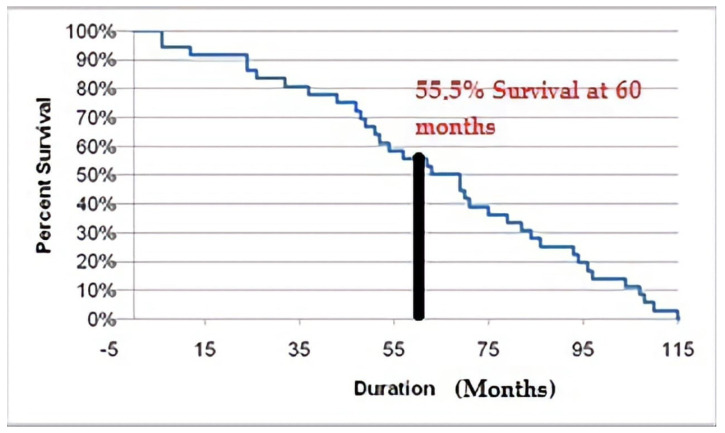
Non-linear analysis of survival over time—55.5% 5-year survival rate in study group.

**Figure 4 diagnostics-13-02069-f004:**
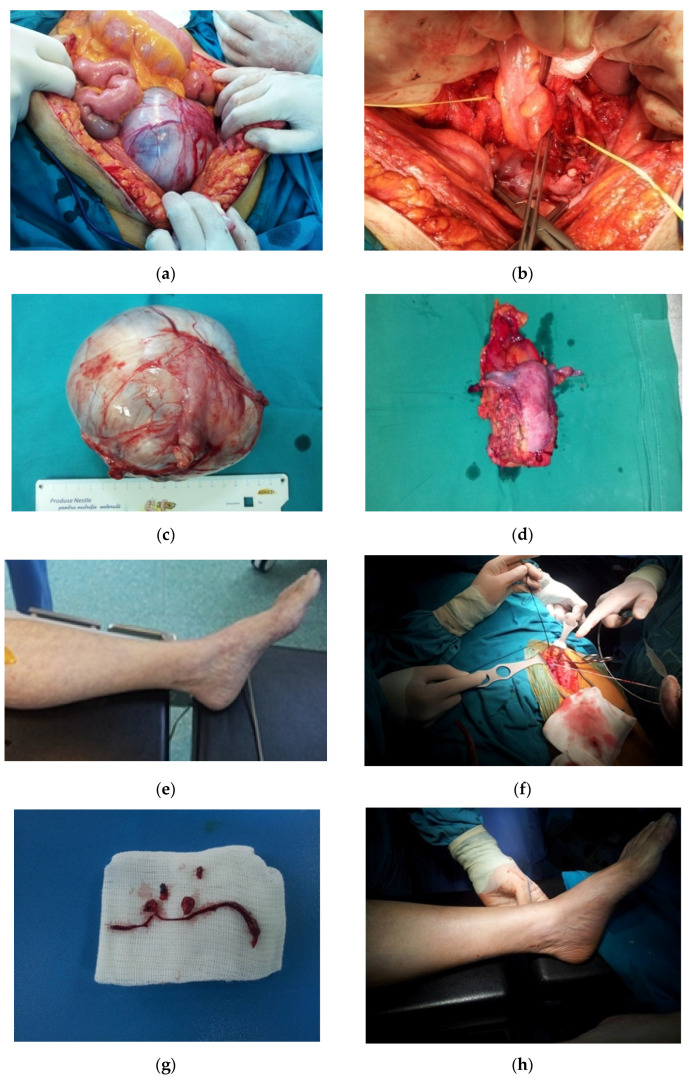
Clinical and intraoperative aspects of patient with synchronous ovarian and cervical invasive cancers, presenting with numerous prior vascular surgical procedures: (**a**) voluminous ovarian mass evident immediate after laparotomy; (**b**) intraoperative aspect—anterior pelvectomy; bilateral Dacron aorto-femoral prostheses can be observed crossing the ureters; the ureters are highlighted by the yellow tractors; (**c**) 17/14 cm ovarian mass after excision; (**d**) anterior pelvectomy resection specimen; (**e**) acute ischemic syndrome of left lower limb, which developed on the 3rd day post surgery; (**f**) intraoperative aspect of vascular reintervention—removal of the thrombus in the prosthetic left branch of the deep femoral artery via the femoral pathway using a Fogarthy probe; (**g**) removed thrombi; (**h**) resolution of acute ischemic syndrome of left lower limb after vascular reintervention.

**Table 1 diagnostics-13-02069-t001:** Histopathological aspects of the synchronous primary invasive gynecological cancers.

Primary Uterine Cancer	Histology of Uterine Cancer	Secondary Neoplasm	Histology of Secondary Invasive Cancer
Endometrial (*n* = 20)	Endometrioid (*n* = 14) Adenocarcinoma (*n* = 3) Stromal sarcoma (*n* = 1) Clear-cell carcinoma (*n* = 1) Serous carcinoma (*n* = 1)	Ovarian (*n* = 6)	Serous carcinoma (*n* = 2) Endometrioid (*n* = 3) Adenocarcinoma (*n* = 1)
		Vaginal (*n* = 2)	Adenosquamous carcinoma (*n* = 2)
		Cervical (*n* = 12)	Keratinized squamous carcinoma (*n* = 2) Non-keratinized squamous cell carcinoma (*n* = 8) Endocervical adenocarcinoma (*n* = 2)
		In situ (*n* = 43)	
Cervical (*n* = 16)	Keratinized squamous carcinoma (*n* = 3) Non-keratinized squamous-cell carcinoma (*n* = 12) Endocervical adenocarcinoma (*n* = 1)	Ovarian (*n* = 10)	Serous carcinoma (*n* = 6) Endometrioid (*n* = 2) Adenocarcinoma (*n* = 2)
		Vaginal (*n* = 4)	Adenosquamous carcinoma (*n* = 4)
		Salpinx (*n* = 2)	Serous carcinoma (*n* = 2)
		Bladder (*n* = 5)	Urotelial (*n* = 5)
		Rectum (*n* = 7)	Adenocarcinoma (*n* = 7)
		In situ (*n* = 24)	

**Table 2 diagnostics-13-02069-t002:** Initial pathologic stadialization of SGN cases in study group.

	Uterine Cancer	Stage	Secondary Cancer	Stage
1	Cervical	IIb	Ovarian	IIa
2	Cervical	IIb	Ovarian	IIIc
3	Cervical	IVa	Ovarian	Ib
4	Cervical	Ib	Ovarian	IIIb
5	Cervical	Ib	Ovarian	IIa
6	Cervical	Ib	Salpinx	Ia
7	Cervical	IIa	Vaginal	Ia
8	Cervical	IVa	Vaginal	II
9	Cervical	Ib	Ovarian	Ib
10	Cervical	IIa	Ovarian	Ic
11	Cervical	IIa	Ovarian	IIIb
12	Cervical	IIb	Ovarian	Ib
13	Cervical	IIa	Ovarian	II
14	Cervical	IIb	Salpinx	Ia
15	Cervical	Ib	Vaginal	Ia
16	Cervical	IIa	Vaginal	II
17	Endometrial	IIa	Cervical	Ia
18	Endometrial	IIb	Cervical	Ia
19	Endometrial	IIb	Cervical	IIb
20	Endometrial	IIIc	Cervical	Ib
21	Endometrial	II	Cervical	Ib
22	Endometrial	Ib	Cervical	Ia
23	Endometrial	Ib	Ovarian	IIIb
24	Endometrial	II	Ovarian	Ib
25	Endometrial	Ib	Ovarian	Ia
26	Endometrial	II	Vaginal	Ia
27	Endometrial	II	Ovarian	II
28	Endometrial	Ib	Ovarian	IIIc
29	Endometrial	II	Ovarian	Ic
30	Endometrial	Ia	Vaginal	II
31	Endometrial	II	Cervical	IIa
32	Endometrial	Ia	Cervical	IIa
33	Endometrial	Ia	Cervical	Ib
34	Endometrial	Ib	Cervical	Ia
35	Endometrial	Ia	Cervical	Ib
36	Endometrial	Ib	Cervical	Ib

## Data Availability

The data presented in this study are available on request from the corresponding author. The data are not publicly available, being extracted from the medical documents of the patients of the Bucharest Oncological Institute, which are not accessible to the general public.

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
