# Peer review of "A Decade of Therapeutic Challenges in Synchronous Gynecological Cancers from the Bucharest Oncological Institute"

_diagnostics, 2023, doi:10.3390/diagnostics13122069_

Round 1

Reviewer 1 Report

Dear Authors, thank you for submitting this manuscript. It is very difficult to analyze treatment in synchronous cancers. The rarity of this cases makes it possible to observe only in high-volume centers like yours. The manuscript is well prepared and written in easy to read style.

I recommend to publish it as it is.

Author Response

Dear reviewer 1, we can only thank you for the way you understood and appreciated our work and assure you that in the revised form of this manuscript we have considered every suggestion made to us, trying to bring that recommended "polish" to the manuscript. The manuscript has been revised extensively and I am convinced that the improved version will receive the same appreciation from you. Thank you once again.

Reviewer 2 Report

This manuscript described the review of synchronous gynecological cancers treated in the Bucharest Oncological Institute over a decade. This manuscript is well organized, however, there are several points to be improved as shown below:

1. Is the Abstract at line 29 correct? The data is not shown. Furthermore, it is described that SGNs are usually diagnosed in earlier stages and have a better prognosis than metastatic disease at lines 312-313.

2. This manuscript described the review of synchronous gynecological cancers treated in the Bucharest Oncological Institute over a decade, however, the purpose of this study is obscure. It should be specified.

3. On lines 147-149, cervical cancer also shows abnormal vaginal bleeding therefore another reason such as genetic factor will be considered.

 4. How did the authors define the primary cancer and secondary neoplasm in Table 1?

 5. The authors showed the prognosis data in Figure 3, however, it was not discussed.

 6. The description of highly invasive surgery such as pelvic exenteration is too long. It obscures the purpose of this manuscript.

 7. The authors should first conduct a discussion based on the results.

8. In Table 2, the subclassification of the stage is not shown in several cases.

Author Response

Dear Reviewer 2,

First, we, the authors, would like to express how deeply honored we are that our work was submitted to such a rigorous and extremely patient review. We are delighted that you have dedicated your time to an in-depth analysis of our manuscript and we thank you for your suggestions, which allow us to improve our work and produce relevant material. After reviewing our manuscript accordingly with your suggestions, we also found a few additional small mistakes that we were able to rectify in time, and we are grateful for your contribution.

Second, we have addressed each of your concerns as follows:

  1. Is the Abstract at line 29, correct? The data is not shown. Furthermore, it is described that SGNs are usually diagnosed in earlier stages and have a better prognosis than metastatic disease at lines 312-313.

Yes, it is correct. In line 29 we refer to the situation in Romania. Our cases are usually diagnosed in advanced stages than those described in international literature and this is because of the particularities of the Romanian healthcare system, also the beliefs, knowledge, and misconception of Romanian patients regarding disease and prevention or diagnostic methods. The fact that our cases are diagnosed in advanced stages influences the prognosis and treatment options. In lines 312-313 we describe the situation found in international literature by contrast with Romania.

  1. This manuscript described the review of synchronous gynecological cancers treated in the Bucharest Oncological Institute over a decade, however, the purpose of this study is obscure. It should be specified.

Given the fact that Romania has one of the largest rates of cervical-cancer specific incidence and mortality, and this type of cancer counts for about 20% of the patients treated in the Surgery departments of "Prof. Dr. Al. Trestioreanu" Bucharest Oncological Institute, we had the idea that in Romania there are factors that can influence treatment, diagnosis, and prognosis for gynecological cancers. We tried to identify those factors and their effects in the context of a specific sub-set of our gynecological patients. We added lines 19-20 in the abstract for better explaining the purpose of the study.

  1. On lines 147-149, cervical cancer also shows abnormal vaginal bleeding therefore another reason such as genetic factor will be considered.

In lines 147-149 (lines 152-154 of corrected manuscript) we suggested as a potential reason for early-stage diagnosis of endometrial cancers the obvious bleeding, which the patient cannot ignore and which makes her seek out medical attention sooner. By contrast, other gynecological cancers have more discrete symptoms in early stages, being easily ignored by patients and constituting a reason for delayed diagnosis. Even though cervical cancer can present with vaginal bleeding, the bleeding is usually more reduced than in endometrial cancer and can be ignored by patients with low health education.

  1. How did the authors define the primary cancer and secondary neoplasm in Table 1?

The authors define as primary cancer the one that made the patient come to the doctor. The secondary cancer was considered the one that was diagnosed on the resection specimen. We added the explanation on lines 74-77 of the corrected manuscript.

  1. The authors showed the prognosis data in Figure 3, however, it was not discussed.

In lines 169-172 of the corrected manuscript the data about prognosis is discussed, based on a median follow-up of 64 months. The 5-year survival rate is 55.5% and the disease-free survival of 60.2 months. Given the fact that some of our patients had a shorter follow-up we represented the data as a non-linear analysis of survival over time taking into account the observational loss of patients (compensating for patients that had shorter follow-ups and for which there have no certain data about their survival).

  1. The description of highly invasive surgery such as pelvic exenteration is too long. It obscures the purpose of this manuscript.

The part of text in Results referring to the pelvic exenterations does not provide a full description of the procedure and only shows some aspects encountered in our patients. We tried to illustrate the fact that given our cases particularities, some required extensive highly invasive surgery as treatment. The results presented in lines 193-197 and 204-206 of the corrected manuscript are referring to our series of cases presenting the impact such extensive surgery has both on prognosis and on quality of life. In lines 212-237 of the corrected manuscript, we described 2 of our patients that required additional vascular procedures along with the pelvic exenteration, the complications and the ways of solving those cases. This was done in order to better explain the particularities of our cohort.

  1. The authors should first conduct a discussion based on the results.

We opted for a comparative discussion of literature versus our study - our results are discussed on lines 300-317, 329-335.

  1. In Table 2, the subclassification of the stage is not shown in several cases.

We added the complete subclassification of stage in table 2.

Reviewer 3 Report

In my opinion, the analyzed topic is interesting enough to attract the readers’ attention. The goal of this article was to review the synchronous gynecological neoplasias treated in the Bucharest Oncological Institute's surgery departments, over a decade.  I think that the abstract of this article is well organized and clear. In my opinion, the discussion could be studied in depth and extended. Maybe, it could be useful the evaluation of the fraily of these patients in order to reduce the postoperative complications. In particular I suggest these article to get deeper in the topic: The role of preoperative frailty assessment in patients affected by gynecological cancer: a narrative review Ottavia D’Oria, Tullio Golia D’Auge, Ermelinda Baiocco, Cristina Vincenzoni, Emanuela Mancini, Valentina Bruno, Benito Chiofalo, Rosanna Mancari, Riccardo Vizza, Giuseppe Cutillo, Andrea Giannini Vol. 34 (No. 2) 2022 June, 76-83 doi: 10.36129/jog.2022.34. Because of these reasons, the article should be revised and completed. Figures are interesting and tables are clear. Considered all these points, I think it could be of interest for the readers and, in my opinion, it deserves the priority to be published after minor revisions

Author Response

First, we would like to thank you for your kind evaluation of our study.

We subjected our manuscript to extensive grammar check and the text was revised by one of our authors who holds an English degree of Native-speaking level.

We have considered your suggestion about using a frailty scale for our patients. Unfortunately, in our institution there is no standardized frailty scale being applied for surgical patients. We acknowledge this as a limitation of our study and have stated so in the Limitations part of the manuscript, mentioning the benefits that the scale you proposed could bring to our analysis (a higher frailty being associated with lower survival rates). The article you mentioned was cited in our bibliography. In addition, we would like to say that among our patients there were no fatal complications, only 2 patients requiring reinterventions – but this had more to do with the severity of disease and invasiveness of surgery performed, rather than the frailty of the patient.

Round 2

Reviewer 2 Report

 The authors have mostly revised the manuscript according to the reviewer’s comments, however, the revision in terms of prognosis is insufficient.

Although the authors described that SGNs in this study had a more advanced stage and a worse prognosis than previous reports and addressed its reason, it is difficult to tell if it has a better or worse prognosis from Figure 3. It should be addressed.

Author Response

Dear Reviewer 2,

We thank you for your suggestions – they were extremely helpful and revealed an insufficient explanation on our part of the results.

Figure 3 cannot be changed since it shows our survival results. But we added in lines 176 to 179 of the corrected manuscript supplementary discussion of our survival results by contrast with those presented in international literature. By doing so we showed that our survival results are worse than those usually associated with the synchronous gynecological neoplasia. We also presented a possible explanation for our results.

All articles used for the comparison with our results were cited in the References section of our manuscript.